# NHS 'Learning from Deaths' reports: a qualitative and quantitative document analysis of the first year of a countrywide patient safety programme

Zoe Brummell [1,2] Cecilia Vindrola-Padros [1] Dorit Braun,[3]
S Ramani Moonesinghe [1,2,4]

¹Department of Targeted Intervention, Division of Surgery and Interventional Science, University College London, London, UK
²Department of Anaesthesia and Intensive Care Medicine, University College London Hospitals NHS Foundation Trust, London, UK
³Advisor/Lived experience, London, UK
⁴National Institute for Academic Anaesthesia, Royal College of Anaesthetists, Health Services Research Centre, London, UK

**Correspondence to**
Dr Zoe Brummell;
z.brummell@nhs.net

## ABSTRACT

**Objectives** To review how National Health Service (NHS) Secondary Care Trusts (NSCTs) are using the Learning from Deaths (LfDs) programme to learn from and prevent, potentially preventable deaths.

**Introduction** Potentially preventable deaths occur worldwide within healthcare organisations. In England, inconsistencies in how NSCTs reviewed, investigated and shared LfDs, resulted in the introduction of national guidance on 'LfDs' in 2017. This guidance provides a 'framework for identifying, reporting, investigating and LfDs'. Amendments to NHS Quality Account regulations, legally require NSCTs in England to report quantitative and qualitative information relating to patient deaths annually. The programme intended NSCTs would share this learning and take measurable action to prevent future deaths.

**Method** We undertook qualitative and quantitative secondary data, document analysis of all NSCTs LfDs reports within their 2017/2018 Quality Accounts (n=222).

**Results** All statutory elements of LfDs reporting were reported by 98 out of 222 (44%) NSCTs. The percentage of deaths judged more likely than not due to problems in healthcare was between 0% and 13%. The majority of NSCTs (89%) reported lessons learnt; the most common learning theme was poor communication. 106 out of 222 NSCTs (48%) have shared or plan to share the learning within their own organisation. The majority of NSCTs (86%) reported actions taken and 47% discussed or had a plan for assessment of impact. 37 out of 222 NSCTs (17%) mentioned involvement of bereaved families.

**Conclusions** The wide variation in reporting demonstrates that some NSCTs have engaged fully with LfDs, while other NSCTs appear to have disengaged with the programme. This may reveal a disparity in organisational learning and patient safety culture which could result in inequity for bereaved families. Many themes identified from the LfDs reports have previously been identified by national and international reports and inquiries.

## INTRODUCTION

Globally, adverse events while receiving healthcare is a leading cause of morbidity and mortality.[1] The percentage of preventable or potentially preventable deaths is likely to lie somewhere between 0.5% and 8.4% of

### Strengths and limitations of this study

► This is the first study to our knowledge to analyse Learning from Deaths (LfDs) reporting.
► Quality Accounts from all National Health Service Secondary Care Trusts (NSCTs) in England legally required to report LfDs were included in both the quantitative and qualitative analysis. Including all NSCTs in the qualitative analysis has ensured complete and thorough data capture. Despite attempts to minimise inherent researcher bias, the qualitative analysis may have been influenced by to some extent.
► This study has ensured the inclusion of views from bereaved relatives, through patient and public involvement. The authors conclude that these views are essential to improving patient safety.
► This is an analysis of the very first year of LfDs reporting and reports could underrepresent current NSCT engagement in the LfDs process.
► NSCTs may be undertaking elements of the LfDs programme that were not statutory reporting requirements such as family/carer engagement, but not reporting on these as it was not a regulatory requirement.

hospital deaths.[2–6] In England, between April 2017 and end of March 2018, there were 299 000 deaths occurring in hospital or within 30 days of discharge, this amounts to an estimate of between 1495 and 25 116 potentially preventable deaths.[7] There is a moral imperative for healthcare organisations to learn from these deaths and take measurable action to prevent potentially preventable deaths.

Healthcare organisations are made up of individuals who have the ability to learn, however, organisational learning is 'more than the sum of individual learning' and is distinct from unreflective action taking.[8] It is more than simply creating change for change's sake, as an 'illusion of learning'.[9] Organisational learning is the ability to

apply knowledge and understanding to increase effective organisational action.[8 10] Effective organisational learning is crucial to improve patient safety and probably requires both safety-I (understanding why things go wrong) and safety-II (understanding why things go right) approaches.[11 12] Central regulation and performance management may have some effect on improving care, but quality improvement, leadership, public engagement, proper resourcing, education and training are needed for a safer health service.[12]

In April 2016, an independent review demonstrated a lack of systematic approach and meaningful change in response to unexpected deaths at Southern Health National Health Service (NHS) Foundation Trust.[13] The Care Quality Commission (CQC), which is responsible for monitoring, inspection and regulation of healthcare services within England, conducted a wider review into the investigations of deaths. They found inconsistencies in the way NHS Secondary Care Trusts (NSCTs) became aware of, investigated and shared learning from deaths (LfDs).[14] In response, the NHS launched a new programme of work to improve standards. This included national guidance on 'LfDs, providing a framework for NSCTs on 'identifying, reporting, investigating and LfDs in care'. The objectives of the guidance included supporting the NHS in England to develop an understanding of why deaths arising from problems in care occur, with the aim of ensuring that findings are shared and acted on, to prevent recurrence.[15] In July 2017, guidance was published on implementing the LfDs framework at NSCT board level,[16] and amendments to statutory regulations followed. These changes made annual reporting of both quantitative and qualitative information relating to patient deaths a legal requirement in England (NHS quality account regulations 2010 (2017 No.744)).[17] The reporting mechanism was built into the NHS 'Quality Accounts' system—where NSCTs are legally required to produce a publicly available annual report about the quality of their services (UK government legislation).[18]

Guidance was not given on expected number of deaths, how to judge if a death was more likely than not due to problems in care, or on examples of learning, actions or how to assess impact of any actions. It was instead left to individual NSCTs to decide how they would undertake these requirements. Guidance was given that NSCT board leadership should 'shares relevant learning across the organisation and with other services',[15] and that NSCTs should 'engage meaningfully with bereaved families and carers'.[19] It was not a statutory requirement to report on bereaved family and carer engagement or to report sharing of learning. This study analyses if NSCTs are reporting as legally required, evaluates the quality of reporting, and determines whether there is evidence of effective organisational learning, sharing of learning and engagement with bereaved families and carers.

## METHODS

This is a qualitative and quantitative study of an NHS safety improvement programme. We undertook analysis of 2017/2018 Quality Account data from NSCTs in England. We excluded Quality Accounts from ambulance trusts because in 2017/2018 they were not required to report. This study has been reported using Standards for Reporting Qualitative Research.[20]

Our objectives were to describe the quality of reporting, and to thematically analyse the reports to derive key learning for the NHS and beyond. We undertook analysis of LfDs as set out in the 2017 amendment to the NHS 2010 Quality Account regulations.

Our evaluation of the quality of reporting involved review of compliance of reports against regulation numbers 27.1–27.6 (table 1).[17] Where NSCTs did not fully report we sought to understand why this may have been the case from comments within the Quality Account itself.

In addition to statutorily required reporting, we also looked for evidence within the 2017/2018 LfDs report of family/carer engagement, which included evidence of involvement in learning and/or addressing family/carer concerns and/or appointing family liaison officer or similar as a result of a patient death. We also looked for evidence of sharing LfDs incidents both within the NSCT and more widely (eg, with other organisations). Both sharing learning and family/carer engagement were recommended in the LfDs national guidance.[15]

Quantitative analysis of regulation 27.1–27.3 was undertaken and reported using descriptive statistics.

| Table 1 | NHS Quality Accounts LfDs regulations[17] |
|---|---|
| **Regulation no** | **Summary of regulatory requirement** |
| 27.1 | The no of patients who have died during the annual reporting period. |
| 27.2 | The no of the deaths (in 27.1) that have undergone a case record review or investigation. |
| 27.3 | An estimate of the no of deaths in 27.2 which the NSCT judges to be more likely than not to have been due to problems in care, with explanation of method to assess this. |
| 27.4 | What the NSCT has learnt from reviews/investigations in relation to deaths (in 27.3). |
| 27.5 | A description of the actions the NSCT has taken or will take in response to what they have learnt |
| 27.6 | An assessment of the impact of the actions (from 27.5). |

LfDS, Learning from Deaths; NSCT, National Health Service Secondary Care Trusts.

Qualitative analysis to derive key learning themes from regulations 27.4–27.6, sharing learning and family/carer engagement was undertaken through document analysis as described by Bowen[21] using both content and thematic analysis, and through exploratory data analysis.[21 22] Both deductive and inductive approaches were used. We first identified initial LfDs learning and action themes for reporting, and then developed a classification system for these. The first investigator (ZB) reviewed and analysed twenty 2017/2018 quality accounts, undertook open coding (inductive) and combined this with information presented at the NHS Improvement London LfDs Network (October 2018), where themes (mixed learning and action) from London NSCTs were discussed (deductive). Following the initial review, we reviewed the further 202 NSCT 2017/2018 Quality Accounts. Each Account was reviewed by the same reviewer twice to ensure full data capture. Researchers used the process of bracketing to reduce subjective analysis.[23] During data capture further themes emerged, were modified, merged and changed iteratively. Recurring themes were identified using exploratory data analysis,[22] coding, identification of themes, recoding and using frequency charts. Data were captured in Microsoft excel (V.16.15).

### Patient and public involvement

This study forms part of a larger programme of work which is overseen by a public and relatives steering group to improve relevance from the perspective of those affected by deaths in healthcare and to reduce biases from the healthcare staff researchers. The steering group have been involved in the planning, design and development of conclusions, through face-to-face meetings and email correspondence. The involvement of a steering group member in authoring this paper has significantly and positively influenced the reporting of this study, ensuring focus on reporting family involvement. The authors reflect that patient and public involvement (PPI) has been essential to this study to ensure that the views of bereaved family members were central to the concerns examined. The reporting of PPI has been undertaken using guidance for reporting involvement of patients and the public 2—short form.[24]

### RESULTS

Quality Accounts were reviewed for all 222 NSCTs in England.

### Quality of reporting

Ninety-eight out of 222 (44%) NSCTs reported all six statutory elements of the LfD reporting framework. Two NSCTs did not report any parts of the LfDs regulatory requirements.[25 26] The total number of deaths reported (regulation 27.1) varied from 3 deaths to 7756 deaths (median 1210.5, range 7753).[27 28] The number of case record reviews or investigations undertaken relative to the number of patient deaths in individual NSCTs varied between 0.2% and 100% of deaths; the average was 43.7% (median 36.5, range 99.8).

### Number of deaths which the NSCT judges to be more likely than not to have been due to problems in care, with explanation of method used to assess this

There was variation between 0% and 13% in the number of deaths which the NSCT judged to be more likely than not to have been due to problems in care (median 0.2, range 13). Twenty-two NSCTs did not report any figure in this section of the quality accounts, reasons given for this included:

► 'Data collection challenges'.[29]
► 'Unable to provide a reliable figure'.[30]
► 'We do not carry out investigations with a view to determining whether the death was wholly or partly due to problems in the care provided'.[31]
► 'Currently, no research base on this for mental health services and no consistent accepted basis for calculating this data'.[32]

A total of 111 out of 222 NSCTs (50%) noted the use of Structured Judgement Reviews (SJRs) (either Royal College of Physicians or Royal College of Psychiatrists) either alone or in combination with other forms of investigation or review to assess problems in care.[33] NSCTs not using SJRs used a variety of other methods including: Confidential Enquiry into Stillbirths and Deaths in Infancy framework, Root Cause Analysis and PReventable Incidents Survival and Mortality methodology.[34 35]

### Plans for assessment of impact

Regulation 26.6 asked NSCTs to undertake 'an assessment of the impact of the actions'. 105 out of 222 NSCTs (47%) discussed assessment of impact. This includes trusts that had a plan of any sort including a future plan. Several NSCTs used audits and/or quality improvement projects to check that actions are implemented. One NSCT stated 'Many of these actions are difficult to objectively assess in terms of their impact as they may relate to rare occurrences, which are difficult to meaningfully audit'.[36] The 47% of NSCTs who had a plan for assessment of impact does not include NSCTs that acknowledge the need to assess the impact but stated that it was too early to be able to undertake this (or words to this effect).[37 38] Some NSCTs have reported the results of the assessment of impact that they have already undertaken.[39] Several NSCTs appear to have misunderstood, for example, reiterating the purpose of the LfDs programme, instead of assessing impact.[40 41]

### Evidence involvement of family/carers in learning

In the 2017/2018, LfDs reports 37 out of 222 NSCTs (17%) mentioned the involvement of families/carers either in the investigation process or in shared learning or that they communicate with/support/engage/consider families/carers after a patient dies.[42–44] A good example of working with families from one NSCT LfDs report states: 'The Trust continues to learn the importance of communication

with families after a death has occurred and that through meaningful engagement after a death by inviting them to contribute to the terms of reference for investigations a more detailed, meaningful and richer account of the person's care and treatment is realised'.[45] One NSCT LfDs report discusses that as an action undertaken they sought to gain better education and training for staff about the importance of positive family engagement through expert external training.[46] Thirty-eight NSCTs (17%) discussed as an 'action' that they plan to work with/communicate with/engage/support families/carers. Many of these NSCTs are the same NSCTs already undertaking family/carer engagement.

### Evidence learning shared more widely
In the 2017/2018, trust LfDs reports 106 out of 222 NSCTs (48%) have shared or plan to share the learning more widely within their own organisation, through a variety of communication mediums: Face to face meetings or events, intranet (as case studies, safety alerts, newsletters).[36 44 47] Seventeen out of 222 (8%) NSCTs have shared or plan to share the learning outside their organisation, with neighbouring NSCTs or other national organisations.[47–50]

### Key findings from the reports
#### Lessons learnt
Regulation 27.4 asks NSCTs to describe 'what the provider has learnt from reviews/investigations in relation to deaths' where this was related to deaths which the NSCT judged to be more likely than not to have been due to problems in care (regulation 27.3). 25 out of 222 NSCTs (11%) did not report any lessons learnt from deaths; of these 25 NSCTs, 9 NSCTs had reported 1 or more death judged to be more likely than not due to problems in care, the other 16 NSCTs had either reported zero deaths judged to be more likely than not due to problems in care or had not reported. However, 49 out of 222 NSCTs (22%) which reported that they had no deaths judged more likely than not due to problems in care, also reported lessons learnt, many caveating this with an explanation that they had learnt valuable lessons through the process of case note review/investigation. The most common learning themes from all NSCTs who reported learning can be found in table 2. An overview of the themes arising can be found in the frequency table (figure 1).

Some NSCTs have undertaken analysis of their learning and described common themes.[51] Some have gone into great detail.[52] Others have described a specific case or cases.[53] Some NSCTs have identified learning and actions together, without differentiating the learning from the action. The lack of structure in reporting makes it difficult to always understand exactly what the problem was leading to the learning.[54] Some NSCTs identified 'Good practice' as learning points.[55] Occasionally NSCTs did not necessarily learn from patient deaths, but from the overall LfDs process.[56]

### Actions taken or planned to be taken
NSCTs were asked to undertake 'a description of the actions the NSCT has taken or will take in response to what they have learnt'. Thirty out of the 222 NSCTs (14%) did not report any actions taken as a result of learning. One reported that they felt they were 'at too early a stage of development to be able to take actions from specific learning'.[37] The most common action themes from all NSCTs who reported actions can be found in table 3. An overview of the themes arising can be found in the frequency table (figure 2).

The level of detail with regards to actions taken varies greatly with some NSCTs listing some specific actions as bullet points.[39] Others have described a specific case or cases.[57 58]

### DISCUSSION
This study demonstrates wide variation in both the quality of reporting and the findings from LfDs reports. Considering this is a new programme, introduced part-way through 2017/2018, with limited guidance, the overall findings are somewhat encouraging. Nearly all NSCTs reported at least one or more element of the statutory LfD requirements. Most NSCTs reported lessons learnt and/or actions taken, while less than half discussed assessment of impact. The lessons learnt were varied. The most common learning theme reported was poor communication, with the most common action theme reported being; review of process/standard operating procedure/pathway.

### Quality of reporting
Reporting variation may be due to differences in interpretation of the guidance and statutory requirements. There is

| Table 2 | The five most common learning themes across all NSCTs | |
| --- | --- |
| **Learning themes** | **No of NSCTs citing theme, (%)** |
| Poor communication (including language barrier and problems with handover) | 90 (46) |
| Problem in recognition and escalation of deteriorating patients | 83 (42) |
| End of life planning or treatment escalation planning not evident/incomplete | 82 (42) |
| Problems with documentation including consent, details patient team and NOK | 80 (41) |
| Lack of clinical knowledge, consideration differential/delay diagnosis or seeking advise | 53 (27) |

NOK, next of kin; NSCTs, National Health Service Secondary Care Trusts.

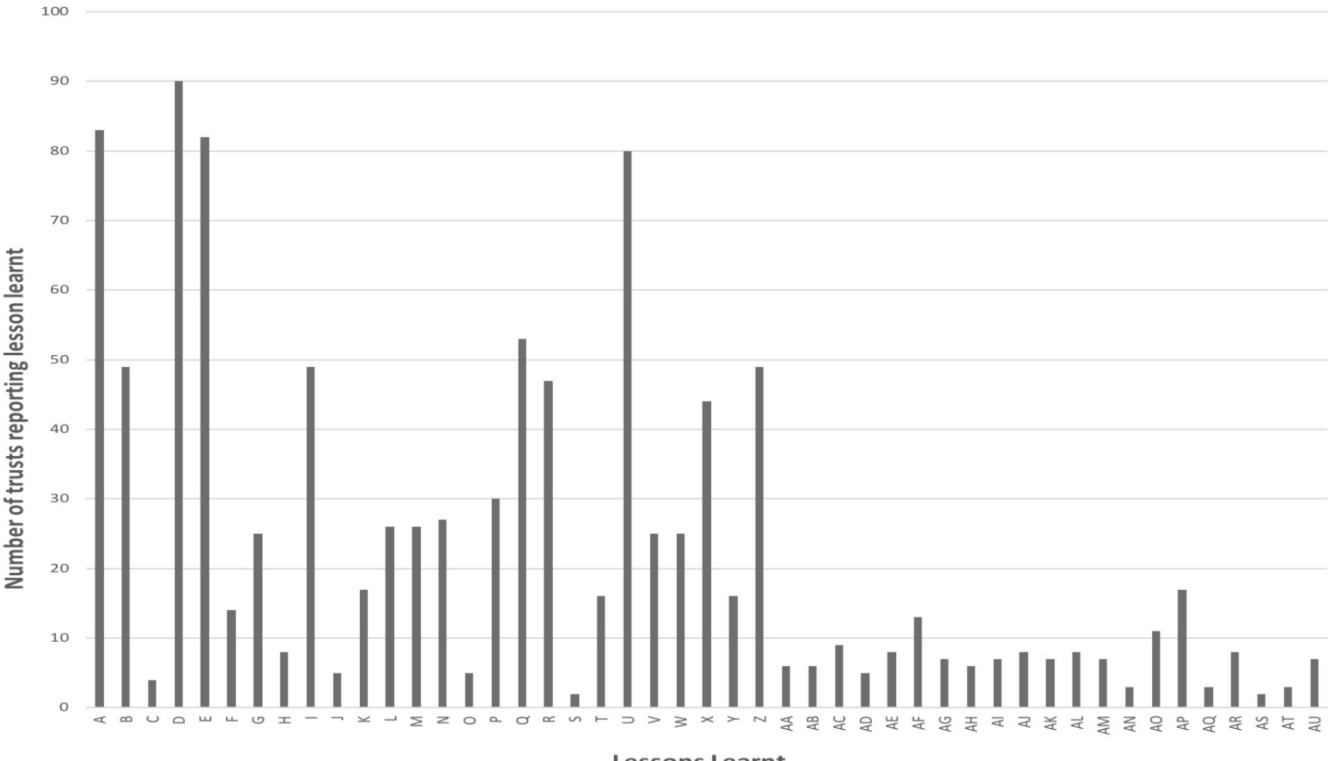

**Figure 1** Frequency table of lessons learnt (all NSCTs n=222). (A) Problem in recognition and escalation of deteriorating patients (B) lack of or awareness of or following protocol/guideline/bundle. (C) Problem in assessment or experience related to learning disabilities. (D) Poor communication (including language barrier and problems with handover). (E) Problem with end-of-life planning or treatment escalation planning. (F) Problem with death certification or confirming death. (G) Problem with discharge (timing/letters/delay/information for patients). (H) Difficulty accessing support services/ out of hours services/ specialist services. (I) Problem/lack of risk assessment/interventions. (J) Lack of knowledge of hospital layout/equipment. (K) Problem with patient transfers. (L) Problem assessing/providing nutrition/fluids/electrolytes. (M) Lack of senior/consultant review, input, planning. (N) Excellent/good care/management. (O) Prompt senior review. (P) Good communication/collaboration/ teamwork. (Q) Lack of clinical knowledge, consideration differential diagnosis or seeking advise. (R) Problem with/lack of prescribing or side-effects or administration of medications. (S) Problem with 'duty of Candour' (T) delay to acting on results. (U) Problems with documentation including consent. (V) Delay/problem in requesting or interpretation of investigations. (W) Lack of/problem with monitoring/observations/recording. (X) Lack of/or problem with sharing information with other providers/ services/specialties. (Y) Delay in reviewing patient (Z) delay in treatment/incomplete management including care plans and pain management. (AA) Poor continuity of care/team work. (AB) Concerns with prehospital care (residential settings/wider societal issues). (AC) Lack of familiarity with or standardisation or availability of equipment. (AD) Problem related to workforce or staffing or supervision of staff. (AE) Misfiled documents/lost notes/problems in storage or access of notes/scans. (AF) Problem with recognition/management of acute kidney injury. (AG) Lack of multidisciplinary team involvement/discussion/decision. (AH) Problem with competency or complication in undertaking procedure/operation. (AI) Problem related to infection control. (AJ) Lack of/problem with assessment of mental health needs and/or follow-up. (AK) Problem related to appropriateness of patient ward allocation or relocation. (AL) Problem with preoperative assessment/perioperative management. (AM) Problem with capacity/flow/hospital of department pressures (including A, E). (AN) Deviation from treatment plan or plan not linked with clinical record. (AO) Follow-up planning not evident or incomplete/problem with follow-up. (AP) Problem related to management of physical health problem in mental health setting. (AQ) Problem due to patient not wanting to/unable to engage with treatment (with capacity). (AR) Problem after death (related to postmortem/forensic services or investigation). (AS) Problem with the recognition/management of drug/alcohol withdrawal/recovery. (AT) Lack of supervision or safe accommodation for vulnerable patient (AU) Lack of/problem with engagement with/support of families/carers. NHS, National Health Service; NSCTs, NHS Secondary Care Trusts.

no direct financial penalty for an NSCT not reporting some or all elements of the LfDs statutory requirements in their Quality Accounts. However, penalties can arise during CQC inspections, when an assessment of implementation of LfDs is carried out.[59 60]

The different approaches taken by NSCTs and the heterogeneity of data makes comparison difficult. The variation in the percentage of deaths being reviewed/investigated may be due to some NSCTs not having the capacity to review/ investigate cases, collect and/or report accurately. NSCTs with a very small number of deaths may find it easier to review all deaths than very large NSCTs. Some NSCTs have had mortality review processes in place for several years and have already been reviewing/investigating deaths, making implementation of the LfDs process easier since the structure for reviewing cases and personnel required are already

**Table 3** The five most common action themes across all NSCTs

| Action themes | No of NSCTs citing theme, (%) |
|---|---|
| Review of process/standard operating procedure/pathway | 128 (67) |
| Highlight guidelines or protocols/policy use of guideline/policies or protocols/treatment bundle/toolkit | 96 (50) |
| Implementation programme of work/ education/bundle | 96 (50) |
| Quality improvement work or similar | 90 (47) |
| Work to improve communication/ collaboration/shared learning | 62 (32) |

NSCTs, National Health Service Secondary Care Trusts.

in place. Some NSCTs may have felt at risk from negative attention by declaring total numbers of deaths and deaths judged more likely than not due to problems in care. Many NSCTs did, however, report despite the same risk. It is clear from the LfDs reports that several NSCTs, particularly some mental health and community NSCTs, did not feel that the guidance applied to them, however other similar NSCTs were able to comply with reporting. The results could suggest guidance was written with acute NSCTs in mind and perhaps need to be reconsidered for non-acute NSCTs. Similar findings were noted by the CQC in their report 'LfDs: A review of the first year of NHS trusts implementing the national guidance'.[60]

The variation in deaths judged more likely than not due to problems in care is larger than those noted in previous studies.[2–6] It seems unlikely than many NSCTs would experience no deaths judged more likely than not due to problems in care. This could realistically be the case in specialist NSCTs where the absolute number of total deaths is very small, or community NSCTs with no inpatient beds, but seems unlikely in large acute NSCTs. Despite the improbability several acute NSCTs did report zero deaths judged more likely than not due to problems in care. Further work to understand why these NSCTs reported zero deaths should be undertaken.

The element of the statutory LfDs reporting that prompted poor responses from most NSCTs was 'An assessment of the impact of the actions' and describing how they would undertake this. The vast majority of NSCTs have answered this in a vague manner, seemingly through variable interpretation of the regulation. Improvements could be made by issuing further specific guidance in relation to this element of the reporting. Of the NSCTs who did manage to implement actions and assess impact this was often using quality improvement measurements. The use of quality improvement methodology is felt to be an important overall indicator

of quality by the CQC.[61] Guidance on evaluating the impact of interventions is widely available.[62 63]

Collectively within the LfDs reports, there is much learning, some resulting in impactful actions and high-level organisational learning.[8] This learning could potentially be usefully shared across the NHS and internationally. Some NHS NSCTs appear to have disengaged with the programme. This study suggests a lack of shared learning from the LfDs reports particularly between NSCTs and a lack of family engagement, despite NHS guidance.[19] Since the involvement of families and sharing learning were not statutory requirements of LfDs reporting, they may be underrepresented in the LfDs reports, this should be investigated further before any definite conclusions can be drawn about NSCTs engagement in this element of the LfDs guidance. This study does demonstrate an apparent disparity in organisational learning and safety culture, which results in inequity for families/carers. This should be addressed by the Department of Health and Social Care (DHSC) and associated national bodies. Since the oversight bodies which were established to support the programme in its initial stages have now been stood down this seems unlikely to happen.[64 65]

### Key findings from the reports
Overall consistency with regard to identifying, reporting, investigating, LfDs in care and taking action has improved across most NSCTs. The continual process of learning, action and reflection which characterises effective organisational learning is essential to ensure the change necessary for safer healthcare. This can only be achieved where information and knowledge affecting patient safety is easily accessible to all members of healthcare staff, supporting an overall safety culture.[10 66]

Only a small number of NSCTs did not report any learning, suggesting that most NSCTs were able to engage with this aspect of reporting. Many NSCTs have effectively described lessons learnt and actions taken. However, most of the LfDs report recommendations or actions are fairly non-specific; further detail of actions and their measurable impact would be helpful.

It is of concern that the majority of these lessons and recommended actions have previously been identified in national and international reports and inquiries, looking at the problems associated with preventable deaths. Similar problems found in this study are also highlighted in these reports; poor clinical monitoring, poor recognition of the deteriorating patient, diagnostic errors, poor communication, lack of end of life planning, lack of information sharing between services, inadequate drug and fluid management.[67–75] This suggests many of the same problems reoccur and that healthcare systems do not learn from previous failings and adds weight to the proposition that the NHS as a whole cannot become a learning organisation.[76] In view of this, it is reasonable to question whether the learning

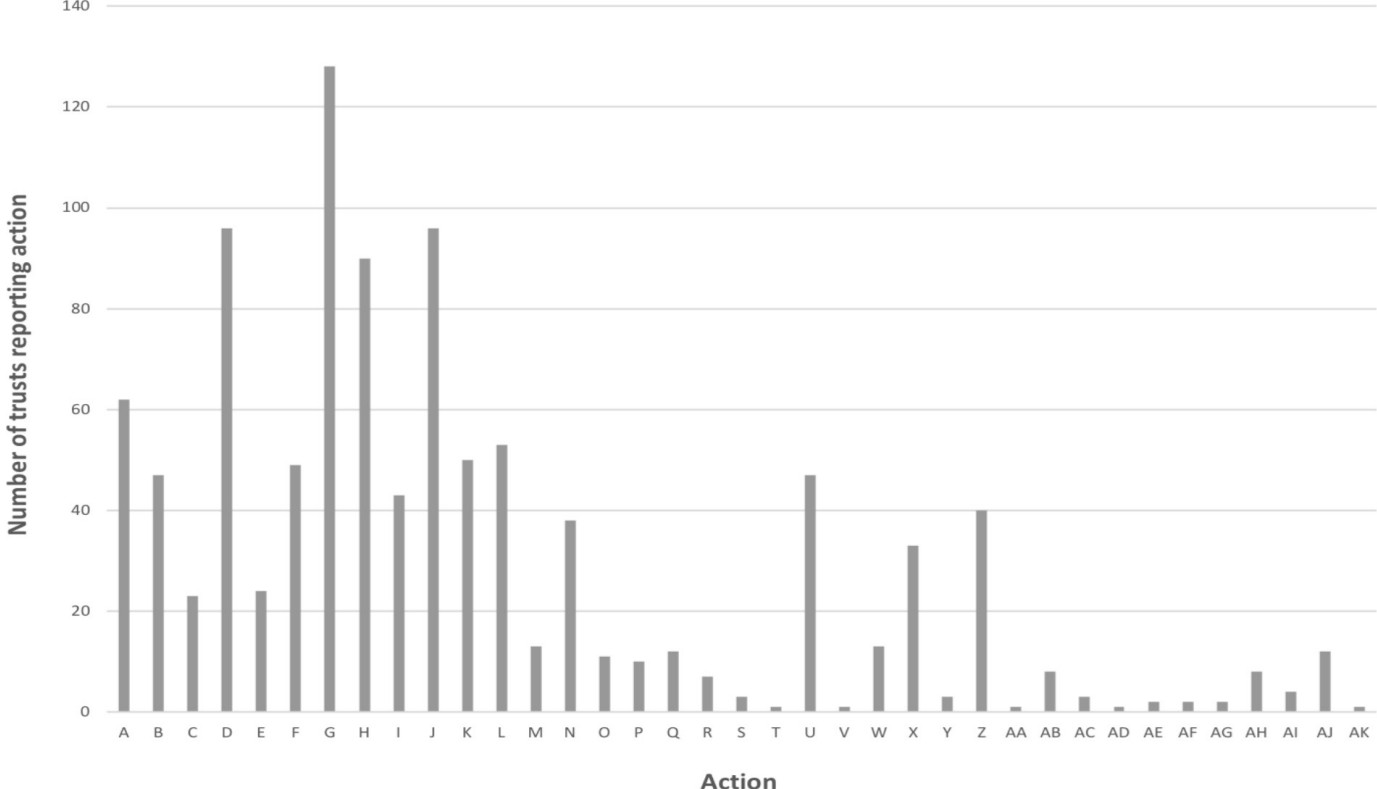

**Figure 2** Frequency table of actions taken (all NSCTs n=222). (A) Work to improve communication/collaboration/shared learning. (B) Improved end-of-life planning (including communication). (C) Improved effectiveness of handover. (D) Highlight or new or use of guidelines/protocols/policy/protocol/treatment bundle/toolkits. (E) Improved mortality review process. (F) Undertake or improve risk assessment/governance process/reporting system. (G) Review of process/SOP/pathway/audit process. (H) Quality improvement work or similar. (I) 'Raising awareness' or 'importance of' or 'reflecting on' (not qualified). (J) Implementation of a programme of work or education (including simulation and induction). (K) Raising awareness (with specific example—'nursing dashboard', 'case presentation'). (L) Use of technology (eg, electronic recording of observations). (M) Rota adjusted to provide better cover or extra lists/sessions. (N) Working/communicating with/supporting families (not end-of-life planning) (O) 'More effective', 'continued efforts', 'seeking advice' 'review/introduce' (not qualified). (P) Solution involving medical examiner role. (Q) Improved senior/consultant involvement (with specific examples). (R) External or internal (peer review) mortality/governance review or investigation. (S) Identification of high-risk patients early. (T) Extend postoperative recovery monitoring. (U) Improved documentation/coding. (V) Follow-up of action plans. (W) Plan to improve sharing of learning. (X) Ensure early warning system in place/used correctly. (Y) Improvement of results reporting and acknowledgement process/ archiving results/scans. (Z) Multidisciplinary team/programme of work setup to address specific problem. (AA) Seek-out and follow expert advise. (AB) Improve review methodology (Such as Structured Judgement Review training). (AC) Develop regional Learning from Deaths network or similar. (AD) Negotiate with coroner for earlier postmortem reports. (AE) Increase emergency operating capability (additional emergency theatre availability). (AF) Improvement to bereavement facilities. (AG) Improved infection control measures. (AH) Supervision discussions/support/feedback for those involved in incidents. (AI) Improved cross-specialty collaboration. (AJ) Increased specialist equipment availability or specialist teams or specialist roles. (AK) Increased engagement in LeDeR process. NHS, National Health Service; NSCTs, NHS Secondary Care Trusts.

arising from LfDs reporting will result in meaningful change. If LfDs findings and recommendations are not implemented, systemic redundancy in the initiative is implied. While individual healthcare practitioners do need to take some responsibility, NSCTs and the DHSC should look at systems, such as institutional account-ability and LfDs programme oversight to optimise outcomes and minimise the risk of fatal patient safety incidents occurring. This lack of change adds to the growing body of evidence suggesting that traditional approaches to organisational learning in healthcare, such as learning from when things go wrong (safety-I) have limited effect and may suggest a role for increased

learning from the patients who have experienced excel-lent patient care and outcome despite being seriously unwell (safety-II).[77]

### Recommendations
In view of the findings from this study, in order to improve reporting quality, our recommendations are as follows:
► A more structured LfDs reporting template, including all regulatory requirements should be implemented through the Quality Accounts.
► NHSE/I-specific guidance should be developed on how NSCTs can undertake 'an assessment of the impact of the actions'.

► To reinstate LfDs robust regulatory reporting oversight in addition to CQC inspections.

In order to improve 'learning and action' from deaths, our recommendations are:

► Annual collection and collation of all NSCT LfDs reporting that is made publicly available.
► Further investigation into how NSCTs currently involve bereaved families and carers.
► Investment in leadership and support for NHS staff to enable a safety culture.

### Methodological limitations

This is an analysis of the very first year of LfDs reporting and reports could underrepresent current NSCT engagement in the LfDs process. NSCTs may be undertaking elements of the LfDs programme that were not statutory reporting requirements such as family/carer engagement, but not reporting on these as it was not a regulatory requirement. NSCT LfDs reports were not created for research analysis and are not standardised, this heterogeneity and subjectivity within the reports reduces equitable comparison.

Despite attempts to minimise inherent researcher bias, such as through PPI involvement and the process of bracketing, the qualitative analysis may have been influenced to a limited extent.

### CONCLUSION

Organisations are variably reporting against LfDs regulations, with overall improved consistency in the way that NSCTs identify, report, investigate and learn from deaths in care since the CQC review.[14] However, more could be done to enhance and strengthen the programme impact, and to assess whether LfDs reporting reflects NSCT LfDs engagement.

On the basis of findings from the 2017/2018 LfDs reports, national programmes led by multidisciplinary healthcare practitioners should be developed to tackle the most common problems which may have contributed to patient deaths. In the first instance programmes tackling the following issues should be developed or strengthened:

► Improving communication.[78]
► Involvement of families in care and in learning.[79]
► Processes to share learning (locally and nationally).[80]

Further work is needed to understand which actions taken by NSCTs result in the biggest impact and for this learning to be shared. While LfDs can be difficult and emotive it is fundamental that healthcare systems ensure learning and impactful change occur.

**Acknowledgements** We would like to acknowledge the work of the Learning from Deaths: Learning and Action (LfDLaA) Public and Relatives steering group in this research. Patient and public involvement in this research was supported by the NIHR UCL Biomedical Research Centre.

**Contributors** ZB conceived and designed the study, undertook analysis, interpretation of the data and drafted the manuscript. CV-P provided critical input to the design, analysis, interpretation of the data and revised the manuscript critically.

DB provided critical input to the design, analysis, interpretation of the data and revised the manuscript critically. SRM provided critical input to the design, analysis, interpretation of the data and revised the manuscript critically. The corresponding author attests that all listed authors meet authorship criteria and that no others meeting the criteria have been omitted. ZB is the guarantor.

**Funding** This study was supported by the National Institute of Health Research (NIHR) University College London (UCL) Biomedical Research Centre Patient and Public Involvement starter funding (BRC617/PPI/ZB/104990).

**Competing interests** ZB worked at NHS Improvement in the medical directorate from August 2017 to August 2018, during which time she undertook some work on the LfDs programme. SRM is the National Clinical Director for Critical and Perioperative Care for NHS England/NHS Improvement, SRM has no link with the LfDs programme.

**Patient and public involvement** Patients and/or the public were involved in the design, or conduct, or reporting, or dissemination plans of this research. Refer to the Methods section for further details.

**Patient consent for publication** Not required.

**Provenance and peer review** Not commissioned; externally peer reviewed.

**Data availability statement** Data are available in a public, open access repository. All the data for this study are publicly available from the 2017/2018 NHS Quality Accounts.

**ORCID iDs**
Zoe Brummell http://orcid.org/0000-0002-6296-5049
Cecilia Vindrola-Padros http://orcid.org/0000-0001-7859-1646
S Ramani Moonesinghe http://orcid.org/0000-0002-6730-5824

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
