## [Reviewer comments · BMJ Open]

ARTICLE DETAILS

TITLE (PROVISIONAL)	NHS 'Learning from Deaths' reports: A qualitative and quantitative document analysis of the first year of a countrywide patient safety programme
AUTHORS	Brummell, Zoe; Vindrola-Padros, Cecilia; Braun, Dorit; Moonesinghe, S. Ramani

VERSION 1 – REVIEW

REVIEWER	A Carson-Stevens Cardiff University
REVIEW RETURNED	12-Dec-2020

GENERAL COMMENTS	Thank you for the opportunity to review this manuscript. Brummell and colleagues have carried out important research to audit the Learning from deaths (LfDs) in the 2017/18 Quality Accounts submitted by all NHS secondary care trusts in England. It is worth pointing out at the outset that they were carrying out the review following the first year of the LfD programme and the authors appropriately recognise that trusts might still be getting to grips with the process and that their results should be reviewed with caution. In light of this, I would side with the authors when they say in their discussion "Considering this is a new programme, introduced part-way through 2017/2018, with limited guidance, the overall findings are somewhat encouraging". That said, I am sure a family member of a patient that has died in England between 2017 and 2018 would struggle to read about the sobering issues highlighted by the authors. If we flip the percentages presented in the abstract alone, 11% did not report lessons learnt, 52% not sharing plans and learning within their organisations, 83% do not involve bereaved families. From their evaluation, organisations can do better and it will be important to carry out a follow-up study to see if improvements have been made by each organisation (and collectively) over time. In some ways, the observations described here, represent some form of indicator about patient safety and just culture / openness in organisations. The methods section will benefit from a clearer position about the study design more detail in parts. Their PPI involvement, benefitting from the support available to a larger programme of work, is very good. The detailed description of the results is a credit to all the hard work arising from forensically examining these Quality Account reports. I think they have done a really nice job at synthesising key points, and notably specifics about the variations in how different trusts demonstrated attainment of the regulation. A consistent reporting on
---

proportions (for dichotomous variables) and medians and ranges (for categories / options) would improve the readability of results. Finally, a more holistic assessment of the strengths and limitations of this study is needed.

I hope the authors find my more detailed review comments below helpful to improve the quality of their manuscript describing this important work.

Introduction

1. In the opening sentence, please amend 'medical treatment' to the term 'healthcare'. Patient safety incidents arise in complex systems for a myriad of reasons and it is not always due to 'treatment' nor the involvement of 'medics' exclusively.

2. Start a new paragraph from "Healthcare organisations are made up of..."

3. Line 38-39: syntax issue – consider amending from "why deaths contributed to by problems in care happen" to "why deaths arising from problems in care occur"

4. Para 2 is helpfully very detailed but to aid readability it could be divided up from where you say "Guidance was not given on..." – consider starting a new paragraph here but do revise the opening frame slightly so it is clear to the reader you are now going to describe the specifics of what support / guidance the trusts were given (or not).

5. Line 56-58, revise to "Given the lack of consistency that led to establishing the..."

6. Where you say, "and to understand if organisational learning in its truest sense is occurring" is unclear / non-specific – could this be revised to something like "to determine whether there is evidence of organisational learning apparent explicitly stated in NHS England Quality Accounts"

7. Overall, a clearer aims / objectives statement is needed. "sets out to analyse LfDs reporting" would be an appropriate way of pitching this to an audience in a colloquial way but falls short of the standards needed for reporting in an academic journal.

Methods

8. Where you say "This is a qualitative and quantitative study...", I was wondering where in the manuscript you were using numerical data to evaluate this programme. This is a secondary analysis of NHS Quality Accounts produced by organisations to examine how they have adopted the principles and mandatory requirements expected following the implementation of the LfD programme. You are using a qualitative study design [content analysis] to analyse Quality Accounts which produced mainly quantitative data and in doing so you have used descriptive statistics to summarise what you observed. It is unclear how you then identified themes – was this from a more in-depth interpretive analysis of a smaller sub-sample, for example (for qualitative detail) therefore generating breadth / depth of insight?

9. Amend "We excluded ambulance trusts (they are not required..." to "We excluded Quality Accounts from ambulance trusts because at the time of our analysis they were not required to report."

10. Whilst it is positive the authors have included reference to O'Brien et al. Standards for Reporting Qualitative Research and included a visual of the standards as a supplementary file. For transparency purposes, it would be helpful if they could explicitly highlight against each standard where and how in the manuscript

	they have addressed this or not. 11. Line 17-18, be consistent and capitalise the Q and A in Quality Account. This is a recurring problem throughout the manuscript which should be addressed because it is the name of the document central to your work. 12. It is unclear what you mean by "Data not found from the trust 2017/2018 Quality Account was not included in the analysis" – do you actually mean, "We noted apparent omissions for each of the Regulations for each secondary care trust's Quality Account report.". 13. Where you say "In addition to statutorily required reporting..." make it clear you looked for that evidence because it was included in the guidance sent to trusts [mentioned in the introduction]. 14. More description is needed about the content analysis. Given the authors used an initial classification to inform a review of LfDs learning and action themes for reporting – was this more a Framework Analysis. At times, there is a subtle difference between these with the former being more inductive in nature. I am not persuaded either way due to lack of detail. Further, the statement "Quantitative analysis was undertaken and reported using descriptive statistics." is made prior to any explanation of how the data were analysed – this is really putting the cart before the horse. Please expand and provide more detail in your description of what you did and be clear on which qualitative study design and method(s) were used. That said, the detail about the initial review, followed by the remaining reports and double-coding is helpful content. 15. State the version of Microsoft Excel used. 16. Please include a statement about research ethics. Results 17. Line 22-24: State the median and then the range in parentheses where you say "There was variation between 0 and 13% in the number of deaths..."; do this consistently, where relevant, elsewhere in results. 18. You helpfully signpost the relevant published Quality Accounts so it is clear to the reader which organisations the finding applies. You could delineate between this list of Quality Accounts (as you might, for example, when you are describing studies included in a systematic review) and tabulate these separately in order to not clutter your main reference list for the paper. You would need to adopt a roman numeral (or similar) labelling system so this was clear though. 19. In the line which starts "Trusts not using SJRs...", can the reader confidently assume trusts cited in references 34 and 35 both used CESDI and RCA and PRISM methodologies? Or did 34 do say CESDI and RCA, whilst 35 did RCA and PRISM? 20. Check journal style, but generally write numbers <10 in full, e.g. "Table 2. The five most" change to "Table 2. The five most..." (the obvious exception is the numbering of tables and figures) 21. The themes described in the Tables 2 and 3 are very clear and read as standalone statements that do not required explanation or an operational definition. 22. Give numbers and percentages in Table 2 and 3. Discussion 23. Syntax issue in the sentence "The most common learning theme was..." 24. Consider starting "A penalty arises..." with "However, penalties can arise during CQC inspections when an assessment of the implementation of LfD is carried out."
--	---

	25. Consider reviewing, and citing, Dr Maria Panagioti's systematic review from BMJ in 2019 (https://www.bmj.com/content/366/bmj.l4185.abstract) where you say "The variation in deaths judged more likely than not..." 26. A more holistic reflection of the strengths and limitations of the study is needed. You only just scratch the surface of this when you acknowledge the reasons for the underrepresentation of family involvement and caution the interpretation of that result. 27. Caution where you say "...prevent errors from occurring...." – we can minimise risk by optimising human factors and the related structures / processes in systems. Suggest you amend to "minimise the risk of fatal patient safety incidents occurring". 28. "from the patients who survive against the odds" suggests they have been subject to some form of patient safety incident and the learning arising from their care experience will be of value. What I think / hope you are actually saying is more focus on patient safety incidents more broadly, and not just the deaths, is needed, as well as 'safety-II'. Given WHO released Guidelines for Incident Reporting and Learning Systems in September 2020, this would seem like a logical conclusion. We know from non-healthcare High Reliability Organisations that vast learning can be gained from understanding near-misses / no harm incidents too. 29. Please provide existing evidence of work that has tackled / is tackling 'improving communication', 'patient and family involvement' and 'dissemination of learning at a local and national level' – citation of the latest evidence on interventions / solutions for these issues could be helpful for the reader. Conclusion 30. When you say "This research shows that the LfDs programme has improved the way that NHS trusts identify, report, investigate and learn from deaths in care." – it implies you had a baseline in which to make this judgement against. Does the research really show that "Organisations are variably reporting against...in terms of X, Y and Z in their Quality Accounts". Be careful not to overreach.
--	--

REVIEWER	Nanako Tamiya University of Tsukuba, Health Services Research
REVIEW RETURNED	03-Jan-2021

GENERAL COMMENTS	The authors conducted a quantitative and qualitative analysis to assess how organizations are using the Learning from Deaths (LfDs) program. This is really an important attempt, but I have some concerns which should be addressed. Especially I would like to mention that mixed-method and Qualitative method is not my area and I was not sure this style of writing is acceptable in this journal or not. Some of the points I mentioned are due to the structure of this paper from my point of view as a reviewer with the Quantitative method. I would like to ask editors to judge it. As it is written "In some qualitative traditions, the results and discussion may not have distinct boundaries" in Standards for Reporting Qualitative Research: A Synthesis of Recommendations (https://journals.lww.com/academicmedicine/Fulltext/2014/09000/Standards_for_Reporting_Qualitative_Research__A.21.aspx) These styles may be acceptable but it is true that it is so difficult for me to understand the construct of this paper.
---

	Major Comments 1. Introduction and all For the international readers, ore explanation about “trust” which may be equivalent to “NHS secondary care trust” will be needed to understand well because this is a unit of analysis and important. 2. Methods, p6 l.24 More explanation of Regulation number may be needed. Is it written in some guidelines and how they were defined? It may be written in ref 17 but more explanation is helpful to understand these important categories. 3. Results, p8 l. 25-3 Are they free comments and each of them appeared only in one trust report which was cited? There may be some similar reasons mentioned by different trust. Which one was the most frequent comment? More information may be interesting and helpful. 4. Discussion, p.11, l.29-32. The authors discussed the possibility that the trusts with experiences of mortality review process were able to implement LfDs process more easily. Is it possible to check this hypothesis by comparing the percentage of deaths being reviewed between trusts with different experiences of mortality review process? 5. Discussion. The authors should describe the limitation of the study in the discussion section. 6. Recommendations, p.13, l.17. The authors mentioned, “Annual collection and collation of all trust LfDs reporting for wider sharing”, but the term “wider sharing” seems vague. Please clarify to what extent the authors recommend the reporting should be shared. Minor Comments 7. Strengths and limitations of this study, p.4 Mentioning what “Lfd” and “PPI” stand for will help readers to get the gist of the study before reading the main body of text. 8. Methods, Patient and public involvement, p.7, l.22-24 The authors used the abbreviation “PPI” in l.22, before they described “PPI” means “patient and public involvement” in l.24. 9. Results, Plans for assessment of impact, p.8, l.54-55. The authors stated “Several trusts appear to have misunderstood...”, but the interpretation of the results should be done in the discussion part, not in the result part. 10. Results, Lessons learnt, p.10, l.16. The authors interpreted the result by stating “This could reduce the transferability...”, but the interpretation of the results should be included in the discussion part, not in the result part.
--	--

VERSION 1 – AUTHOR RESPONSE

Reviewer 1 comments

Comment	Action taken
1. In the opening sentence, please amend 'medical treatment' to the term 'healthcare'. Patient safety incidents arise in complex systems for a myriad of reasons and it is not always	Have made change suggested

due to 'treatment' nor the involvement of 'medics' exclusively.	
2. Start a new paragraph from "Healthcare organisations are made up of..."	Have made change suggested
3. Line 38-39: syntax issue – consider amending from "why deaths contributed to by problems in care happen" to "why deaths arising from problems in care occur"	Have made change suggested
4. Para 2 is helpfully very detailed but to aid readability it could be divided up from where you say "Guidance was not given on..." – consider starting a new paragraph here but do revise the opening frame slightly so it is clear to the reader you are now going to describe the specifics of what support / guidance the trusts were given (or not).	Thankyou, have made change suggested.
5. Line 56-58, revise to "Given the lack of consistency that led to establishing the..."	Have made change suggested
6. Where you say, "and to understand if organisational learning in its truest sense is occurring" is unclear / non-specific – could this be revised to something like "to determine whether there is evidence of organisational learning apparent explicitly stated in NHS England Quality Accounts"	Have changed to 'and to determine whether there is evidence of effective organisational learning'.
7. Overall, a clearer aims / objectives statement is needed. "sets out to analyse LfDs reporting" would be an appropriate way of pitching this to an audience in a colloquial way but falls short of the standards needed for reporting in an academic journal.	Have rewritten the study aim at the end of the introduction section to ensure it is more formal and clearer.
8. Where you say "This is a qualitative and quantitative study...", I was wondering where in the manuscript you were using numerical data to evaluate this programme. This is a secondary analysis of NHS Quality Accounts produced by organisations to examine how they have adopted the principles and mandatory requirements expected following the implementation of the LfD programme. You are using a qualitative study design [content analysis] to analyse Quality Accounts which produced mainly quantitative data and in	Numerical data is used when reviewing the number of patients that have died, the number of deaths that have undergone review/investigation and the number of deaths judged more likely than not to have been due to problems in care. The predominant component of the study is qualitative, but quantitative data analysed with

doing so you have used descriptive statistics to summarise what you observed. It is unclear how you then identified themes – was this from a more in-depth interpretive analysis of a smaller sub-sample, for example (for qualitative detail) therefore generating breath / depth of insight?	descriptive statistics is also included. The qualitative analysis was undertaken using document analysis (Bowen GA. 2009. Document Analysis as a Qualitative Research Method, Qualitative Research Journal. 9 (2). 27-40. doi:10.3316/QRJ0902027) I have changed the wording in the method section to reflect this.
9. Amend "We excluded ambulance trusts (they are not required..." to "We excluded Quality Accounts from ambulance trusts because at the time of our analysis they were not required to report."	Have made change suggested
10. Whilst it is positive the authors have included reference to O'Brien et al. Standards for Reporting Qualitative Research and included a visual of the standards as a supplementary file. For transparency purposes, it would be helpful if they could explicitly highlight against each standard where and how in the manuscript they have addressed this or not.	Have commented against each standard in the Standards for Reporting Qualitative Research and included as a supplementary file (also requested by the editor)
11. Line 17-18, be consistent and capitalise the Q and A in Quality Account. This is a recurring problem throughout the manuscript which should be addressed because it is the name of the document central to your work.	Thankyou, have addressed this
12. It is unclear what you mean by "Data not found from the trust 2017/2018 Quality Account was not included in the analysis" – do you actually mean, "We noted apparent omissions for each of the Regulations for each secondary care trust's Quality Account report.".	Some trusts have additionally produced LfDs reports elsewhere for example as part of board papers. These were not a requirement of the LfDs programme, or a statutory requirement, but could be evaluated further (will include this as a limitation of the study). Have deleted this statement as agree it could be confusing.
13. Where you say "In addition to statutorily required reporting..." make it clear you looked for that evidence because it was included in the guidance sent to trusts [mentioned in the introduction].	Have added an additional sentence at the end of this paragraph to make this clear.

14. More description is needed about the content analysis. Given the authors used an initial classification to inform a review of LfDs learning and action themes for reporting – was this more a Framework Analysis. At times, there is a subtle difference between these with the former being more inductive in nature. I am not persuaded either way due to lack of detail. Further, the statement "Quantitative analysis was undertaken and reported using descriptive statistics." is made prior to any explanation of how the data were analysed – this is really putting the cart before the horse. Please expand and provide more detail in your description of what you did and be clear on which qualitative study design and method(s) were used. That said, the detail about the initial review, followed by the remaining reports and double-coding is helpful content.	Have given further explicit information that the document analysis (through content and thematic analysis) as described by Bowen 2009, and exploratory data analysis (as described by Stebbins 2008) were used. And that both deductive and inductive approaches were used. Agree there is some cross over with framework analysis (although this was not the type of analysis used). Have further described exactly where each type of analysis was used (quant or qualitative)
15. State the version of Microsoft Excel used.	Thanks, have addressed this
16. Please include a statement about research ethics.	The statement about ethical approval can be found in the footnotes
17. Line 22-24: State the median and then the range in parentheses where you say "There was variation between 0 and 13% in the number of deaths..."; do this consistently, where relevant, elsewhere in results.	Thanks, have made change suggested
18. You helpfully signpost the relevant published Quality Accounts so it is clear to the reader which organisations the finding applies. You could delineate between this list of Quality Accounts (as you might, for example, when you are describing studies included in a systematic review) and tabulate these separately in order to not clutter your main reference list for the paper. You would need to adopt a roman numeral (or similar) labelling system so this was clear though.	I think it would be too confusing to have two referencing systems in one paper, but understand that the number of references is large and may seem cluttered because of this
19. In the line which starts "Trusts not using SJRs...", can the reader confidently assume trusts cited in references 34 and 35 both used CESDI and RCA and PRISM methodologies? Or did 34 do say CEDSI and RCA, whilst 35 did RCA and PRISM?	The references in 34 and 35 relate to the methodologies CESDI and PRISM not the trusts using them. 34 MBRRACE-UK 2017. Perinatal Confidential Enquiry. Confidential Enquiry into Stillbirths and Deaths in Infancy (CESDI) framework:19-20 35 Hogan H 2014. Preventable,

	Incidents, Survival and Mortality Study 2 (PRISM) Medical Record Review Manual
20. Check journal style, but generally write numbers <10 in full, e.g. "Table 2. The five most" change to "Table 2. The five most..." (the obvious exception is the numbering of tables and figures)	Have changed, thank you
21. The themes described in the Tables 2 and 3 are very clear and read as standalone statements that do not required explanation or an operational definition.	Thank you, I have kept the table labels to ensure absolute clarity.
22. Give numbers and percentages in Table 2 and 3.	Have made changes, thank you
23. Syntax issue in the sentence "The most common learning theme was..."	Have changed, thank you
24. Consider starting "A penalty arises..." with "However, penalties can arise during CQC inspections when an assessment of the implementation of LfD is carried out."	Have made change suggested
25. Consider reviewing, and citing, Dr Maria Panagioti's systematic review from BMJ in 2019 (https://www.bmj.com/content/366/bmj.l4185.abstract) where you say "The variation in deaths judged more likely than not..."	Thank you, I have previously seen this article, which is excellent. Unfortunately in this article Dr Panagioti's group do not differentiate between prolonged, permanent disability and death and this could explained their even greater proportion of patients effected. The other articles I have cited specifically measure death alone as an outcome.
26. A more holistic reflection of the strengths and limitations of the study is needed. You only just scratch the surface of this when you acknowledge the reasons for the underrepresentation of family involvement and caution the interpretation of that result.	We have added to the strengths and limitations section and in addition reflected further on the limitations of the studying in accurately representing family engagement in the LfDs process.

27. Caution where you say "...prevent errors from occurring...." – we can minimise risk by optimising human factors and the related structures / processes in systems. Suggest you amend to "minimise the risk of fatal patient safety incidents occurring".	Have changed, thankyou
28. "from the patients who survive against the odds" suggests they have been subject to some form of patient safety incident and the learning arising from their care experience will be of value. What I think / hope you are actually saying is more focus on patient safety incidents more broadly, and not just the deaths, is needed, as well as 'safety-II'. Given WHO released Guidelines for Incident Reporting and Learning Systems in September 2020, this would seem like a logical conclusion. We know from non-healthcare High Reliability Organisations that vast learning can be gained from understanding near-misses / no harm incidents too.	Thankyou, have changed this to make it clearer
29. Please provide existing evidence of work that has tackled / is tackling 'improving communication', 'patient and family involvement' and 'dissemination of learning at a local and national level' – citation of the latest evidence on interventions / solutions for these issues could be helpful for the reader.	Thankyou, we have provided references for these.
30. When you say "This research shows that the LfDs programme has improved the way that NHS trusts identify, report, investigate and learn from deaths in care." – it implies you had a baseline in which to make this judgement against. Does the research really show that "Organisations are variably reporting against...in terms of X, Y and Z in their Quality Accounts". Be careful not to overreach.	Thankyou, have changed the start of the conclusion to address these concerns.

Reviewer 2 comments

Comment	Action taken
1. Introduction and all For the international readers, ore explanation about “trust” which may be equivalent to “NHS secondary care trust” will be needed to understand well because this is a unit of analysis and important.	To ensure consistency and ease of understanding for international readers we have replaced Hospital/Organisation/provider/trust with NHS Secondary Care trust; abbreviated to NSCT except where

	alternative terms are used as a direct quote
2. Methods, p6 l.24 More explanation of Regulation number may be needed. Is it written in some guidelines and how they were defined? It may be written in ref 17 but more explanation is helpful to understand these important categories.	As described in the introduction: 'The reporting mechanism was built into the NHS "Quality Accounts" system – where NSCTs are legally required to produce a publicly available annual report about the quality of their services'. We have included in parentheses 'therefore forming United Kingdom government legislation' to ensure it is completely clear.
3. Results, p8 l. 25-3 Are they free comments and each of them appeared only in one trust report which was cited? There may be some similar reasons mentioned by different trust. Which one was the most frequent comment? More information may be interesting and helpful.	The comments have been taken from the Quality Accounts of many different NSCT LfDs reports (please see references). Ideally we would include further comments, but are unfortunately limited by the word count.
4. Discussion, p.11, l.29-32. The authors discussed the possibility that the trusts with experiences of mortality review process were able to implement LfDs process more easily. Is it possible to check this hypothesis by comparing the percentage of deaths being reviewed between trusts with different experiences of mortality review process?	This suggestion has arisen from the LfD reports themselves, several trusts discuss that they have been undertaking mortality reviews for many years. However the reporting on experiences of mortality review process is not consistent and it would therefore not be possible to check this hypothesis.
5. Discussion. The authors should describe the limitation of the study in the discussion section.	Thankyou, we have done this.
6. Recommendations, p.13, l.17. The authors mentioned, "Annual collection and collation of all trust LfDs reporting for wider sharing", but the term "wider sharing" seems vague. Please clarify to what extent the authors recommend the reporting should be shared.	Thankyou, I have changed this to 'that is made publicly available'.

7. Strengths and limitations of this study, p.4 Mentioning what “LfD” and “PPI” stand for will help readers to get the gist of the study before reading the main body of text. 8. Methods, Patient and public involvement, p.7, l.22-24 The authors used the abbreviation “PPI” in l.22, before they described “PPI” means “patient and public involvement” in l.24.	The full form of LfDs (Learning from Deaths) is described in the abstract and the introduction section. I have added it to the ‘strengths and limitations of this study’ section. The full form of PPI (Patient and public involvement) is described in the methods section, now in the correct place. Thankyou
9. Results, Plans for assessment of impact, p.8, l.54-55. The authors stated “Several trusts appear to have misunderstood...”, but the interpretation of the results should be done in the discussion part, not in the result part.	Because we are assessing the quality of the reporting in this part of the results section it is the correct place to put the statement about ‘several NSCTs appear to have misunderstood...’. The misinterpretation by trusts is not an interpretation of the results but is felt by the authors to be an accurate finding of the results.
10. Results, Lessons learnt, p.10, l.16. The authors interpreted the result by stating “This could reduce the transferability...”, but the interpretation of the results should be included in the discussion part, not in the result part.	Thankyou, agree this is in the wrong place and have changed this.

VERSION 2 – REVIEW

REVIEWER	A Carson-Stevens Cardiff University
REVIEW RETURNED	26-Mar-2021
GENERAL COMMENTS	The authors have adequately addressed my questions / proposed changes from the first review.
REVIEWER	Nanako Tamiya University of Tsukuba, Health Services Research
REVIEW RETURNED	03-Apr-2021
GENERAL COMMENTS	Comments to the Authors The authors revised the manuscript properly to meet my previous comments on quantitative methods. Only concern left is as

	follows.please add if possible 1. Introduction, p.6, l.44-47. In response to my previous comment, the authors added explanation about NHS “Quality Accounts” system is based on United Kingdom government legislation. However, it would be more helpful if they put the number of article or section which defined each regulation, for readers to refer to the law.
--	--

VERSION 2 – AUTHOR RESPONSE

Reviewer 2 comments

Comment	Action taken
1. Introduction, p6, l.44-47 In response to my previous comment, the authors added explanation about NHS “Quality Accounts” system is based on United Kingdom government legislation. However, it would be more helpful if they put the number of article or section which defined each regulation, for readers to refer to the law.	We have added further information to direct readers to the section defining the regulation (NHS quality account regulations 2010 (2017 No.744)) in the introductory section. The reference for direct access to the regulation is already available (reference number 17). In addition in the methods section readers can refer to the number for each regulation as shown in table 1 (NHS Quality Accounts LfDs Regulations).